# Observation of non-local impedance response in a passive electrical circuit

Xiao Zhang,[1, *] Boxue Zhang,[1] Weihong Zhao,[2] and Ching Hua Lee[3, †]

[1] School of Physics, Sun Yat-sen University, Guangzhou 510275, China.
[2] School of Electronics and Information Technology,
Sun Yat-sen University, Guangzhou 510275, China.
[3] Department of Physics, National University of Singapore, Singapore, 117542.

(Dated: November 16, 2022)

In media with only short-ranged couplings and interactions, it is natural to assume that physical responses must be local. Yet, we discover that this is not necessarily true, even in a system as commonplace as an electric circuit array. This work reports the experimental observation of non-local impedance response in a designed circuit network consisting exclusively of passive elements such as resistors, inductors and capacitors (RLC). Measurements reveal that the removal of boundary connections dramatically affects the two-point impedance between certain distant nodes, even in the absence of any amplification mechanism for the voltage signal. This non-local impedance response is distinct from the reciprocal non-Hermitian skin effect, affecting only selected pairs of nodes even as the circuit Laplacian exhibits universally broken spectral bulk-boundary correspondence. Surprisingly, not only are component parasitic resistances unable to erode the non-local response, but they in fact give rise to novel loss-induced topological modes at sufficiently large system sizes, constituting a new manifestation of the critical non-Hermitian skin effect. Our findings chart a new route towards attaining non-local responses in photonic or electrical metamaterials without involving non-linear, non-local, active or amplificative elements.

## INTRODUCTION

Non-local or action-at-a-distance phenomena reveal deep enigmatic mechanisms behind interesting physics, from the onset of phase transitions [1, 2] to the causality structure of the universe [3–5]. The presence of non-locality is especially intriguing when it emerges unexpectedly from purely local couplings or interactions, since that implies a hidden mechanism that propagates information beyond intrinsic system length scales. Such emergent non-locality has recently attracted much attention in the context of non-Hermitian bulk-boundary correspondence, where a single coupling perturbation can modify the spectral properties and topological states of the entire system [6–38]

Across existing literature on non-Hermitian lattices, the reported non-local behavior can always be intuitively attributed to directed amplification [39]. Non-Hermitian couplings with asymmetric amplitudes in either direction lead to direction-dependent gain/loss, and together give rise to a chain of amplifications that propagates signals non-locally [39], as experimentally

demonstrated in various metamaterial platforms [7, 40–46].

What is interesting and practical though challenging, however, is achieving such non-local signal propagation when there is no amplification at all. In this work, we experimentally achieved this by detecting non-local impedance response in an electrical circuit designed with purely passive and reciprocal RLC components, which would be easily integrated into chips for sensing applications [47–52]. Specifically, we showed that in our circuit, the impedance between two adjacent nodes can be profoundly modified by cutting off a remote connection, no matter how distant, against common intuition. Unlike existing demonstrations of non-local voltage responses with operational amplifiers [53], our setup contains no intrinsic directionality. Its only non-Hermitian components are the resistors, which are neither active nor chiral, and certainly incapable of causing a cascade of amplifications. Exactly how our non-local response is achieved will be explained in the following.

## RESULTS

### Emergent non-locality without directed amplification

To understand how our setup can exhibit non-local responses with purely passive elements, we first review the mechanism of directed amplification mechanism that gives rise to extreme non-local sensitivity, and how this non-locality is preserved even if the amplifications in different directions cancel. Even though we are ultimately concerned with the impedance response, we shall first illustrate how this non-local mechanism is directly observed from the state evolution under generic non-Hermitian Hamiltonians.

Consider the simplest illustrative Hatano-Nelson (HN) model [54] $H_{\mathrm{HN}} = \sum_x [t^+|x+1\rangle\langle x| + t^-|x-1\rangle\langle x|]$, which amplifies and propagates a state by asymmetric factors of $|t^-|$ and $|t^+|$ towards the left and the right. If $|t^-/t^+| < 1$, an arbitrary signal will be amplified by a factor of approximately $|t^-/t^+|^x$ after propagating $x$ states towards the right (Top chain of Fig. 1a Left) [55]. Likewise, $H_{\mathrm{HN}}^\dagger$ would amplify a state by the same factor towards the left (Bottom chain of Fig. 1a Left). This is a classical manifestation of emergent non-locality, since a small input signal can be amplified to yield a very strong output signal arbitrarily far away. Yet, amplification does not always need to accompany non-locality. If the $H_{\mathrm{HN}}$ and $H_{\mathrm{HN}}^\dagger$ chains

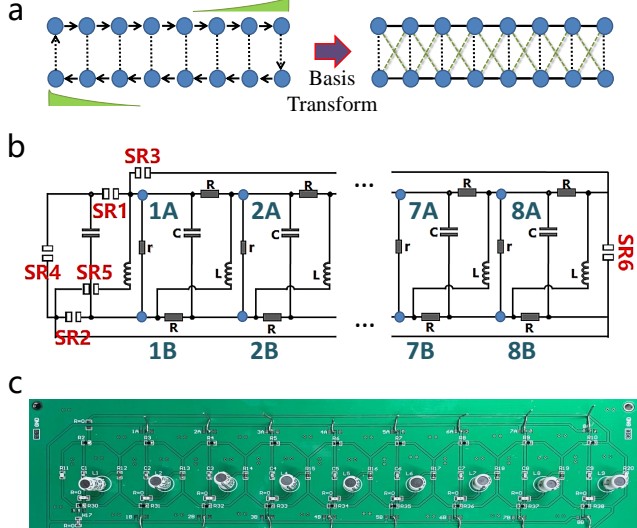

FIG. 1. **Constructing a non-local RLC circuit from the cancellation of non-reciprocity.** (a) Two Hatano-Nelson chains individually experience non-local directed amplification of states (Left). Coupling them can cancel off the directed amplification whilst retaining the non-local response. This inspires the design of our reciprocal lattice with emergent non-locality, which is unitary equivalent to the coupled HN chains (Right). (b) Schematic of our designed reciprocal circuit with 8 unit cells spanned by sublattices $A$ and $B$, as described by $J(k)$ from Eq. 2. By appropriately configuring the switches SR1 to SR6 (see Methods), one can tune the circuit between open and periodic boundary condition (OBC and PBC) configurations. (c) The corresponding experimental circuit, with measured two-point impedances significantly depending on whether PBCs or OBCs are used. Component values are $C = 1nF$, $L = 1mH$, $R = 5k\Omega$ and $r = 50k\Omega$.

interact through a coupling strength $\Delta$ according to

$$H_\Delta = H_{\mathrm{HN}} \otimes |\uparrow\rangle + H_{\mathrm{HN}}^\dagger \otimes |\downarrow\rangle + \mathbb{I} \otimes (|\uparrow\rangle\langle\downarrow| + |\downarrow\rangle\langle\uparrow|)\Delta, \tag{1}$$

the exponentially large amplification of the states near either end would effectively close up the two chains into a "loop" that still experiences the non-local response (Fig. 1a) [55], despite arising from a purely linear system [56]. Importantly, because of the juxtaposition of equal and opposite amplifications, this response is not accompanied by any amplification.

The above-mentioned mechanism for amplification-less non-local response can be adapted to electrical circuits if we consider a circuit Laplacian that is the analog of $H_\Delta$ [57–68]. Unlike a Hamiltonian which represents a time-evolution operator, a Laplacian $J$ describes the steady-state relationship between the input currents $\mathbf{I}$ and electrical potentials $\mathbf{V}$ across the nodes. Explicitly, we write $\mathbf{I} = J\mathbf{V}$, which can be thought of as the matrix form of Kirchhoff's law, with the matrix element $J_{ij}$ describing the linear relationship between the input current at node $i$ and the potential $V_j$ at node $j$.

Since our objective is to design a non-local response circuit that does not even exhibit directed amplifica-

tion, its Laplacian must not explicitly contain $H_{\mathrm{HN}}$ and $H_{\mathrm{HN}}^\dagger$, which harbor asymmetric terms. The most direct way to construct such a circuit Laplacian is to consider a basis-rotated version of $H_\Delta$, such that the asymmetric terms from $H_{\mathrm{HN}}$ and $H_{\mathrm{HN}}^\dagger$ do not exist independently, but are combined to form reciprocal terms (Fig. 1a Right). Physically, this implies that operational amplifiers will not be needed [55], and all non-Hermiticity can be contained in passive lossy resistors. A minimal RLC circuit with such non-locality is illustrated in Fig. 1b, and described by the Laplacian

$$
\begin{aligned}
J(k) &= i\omega C \begin{pmatrix} 1 & -e^{ik} \\ -e^{-ik} & 1 \end{pmatrix} + \frac{1}{i\omega L}\begin{pmatrix} 1 & -e^{-ik} \\ -e^{ik} & 1 \end{pmatrix} \\
&\quad + \frac{1}{r}\begin{pmatrix} 1 & -1 \\ -1 & 1 \end{pmatrix} + \frac{2 - 2\cos k}{R}\begin{pmatrix} 1 & 0 \\ 0 & 1 \end{pmatrix},
\end{aligned}\tag{2}
$$

where $k$ is the quasimomentum along its ladder-like structure. Each unit cell consists of nodes $A$ and $B$ on either side of the ladder, and resistors $r$ and $R$ connect the nodes across and between the rungs respectively. Capacitors $C$ and inductors $L$ connect across the ladder diagonally and induce AC dynamics. At the resonance frequency $\omega = \omega_0 = (LC)^{-1/2}$, its circuit Laplacian simplifies to

$$J(k)|_{\omega=\omega_0} = \frac{2}{R}\left[i(t\sin k)\sigma_y + v(\mathbb{I} - \sigma_x) + (1 - \cos k)\mathbb{I}\right]\tag{3}$$

where $\sigma_x, \sigma_y$ are the Pauli matrices and $t = \omega_0 RC = R\sqrt{\frac{C}{L}}$, $v = \frac{R}{2r}$ are two independent dimensionless control parameters.

To relate this circuit to $H_\Delta$, we perform an unitary basis transformation $U : \sigma_y \to \tilde\sigma_y = U\sigma_y U^{-1} = \sigma_z$ which preserves the spectrum, such that the Laplacian is rotated into the form $J(k)|_{\omega=\omega_0} \to \tilde J(k)|_{\omega=\omega_0}$

$$
\begin{aligned}
&= (2i\omega_0 C\sin k)\sigma_z + r^{-1}(\mathbb{I} - \sigma_x) + 2R^{-1}(1 - \cos k)\mathbb{I} \\
&= \begin{pmatrix} 2\omega_0 C e^{ik} + \xi(k) & -r^{-1} \\ -r^{-1} & 2\omega_0 C e^{-ik} + \xi(k) \end{pmatrix}
\end{aligned}\tag{4}
$$

where $\xi(k) = r^{-1} + 2R^{-1} - 2(R^{-1} + \omega_0 C)\cos k$. In this rotated basis, we evidently have two effective chains coupled by $\Delta = -r^{-1}$, each containing a reciprocal (symmetric) part $\xi(k)$, and an asymmetric part $2\omega_0 C e^{\pm ik} = 2te^{\pm ik}/R$ that supports equal and opposite directed amplification. Expressed in this basis, our Laplacian is clearly a realization of the non-reciprocity-cancellation picture given in Fig. 1a, even though its effective chains contain (fictitious) capacitors with imaginary capacitances $-iC$, and cannot be directly realized without the basis rotation. From it, we can also interpret the dimensionless parameter $v = R/(2r)$ as the effective interchain coupling strength, and the other dimensionless parameter $t = \omega_0 RC$ as controlling the effective coupling asymmetry for the hidden (cancelled) directed amplification.

In the above, what was achieved is the design of a RLC circuit that has similar non-local properties as coupled effective chains with oppositely canceled directed amplification. From its effective model, it is for sure that it exhibits modified spectral bulk-boundary

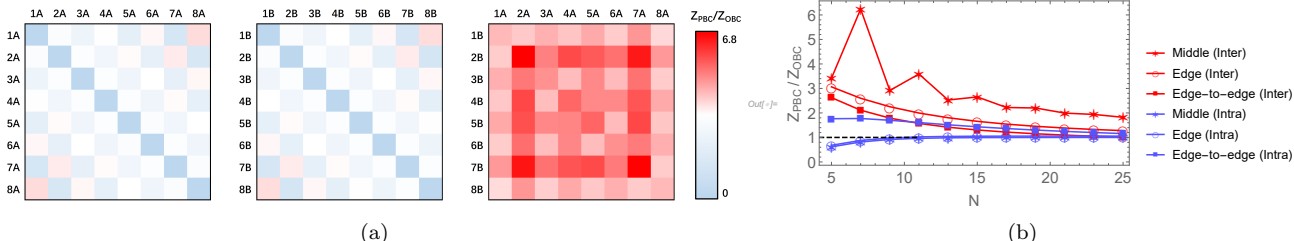

FIG. 2. **Selective non-local impedance response and its persistence in the large system limit.** (a) The 2-point impedance measured across all pairs of nodes of our $N = 8$-unit cell circuit, plotted as a ratio $Z_{\mathrm{PBC}}/Z_{\mathrm{OBC}}$ between PBC and OBC scenarios. Within either the A or B ladders (left two plots), $Z_{\mathrm{PBC}}/Z_{\mathrm{OBC}} \approx 1$ (light blue), implying no nontrivial effect from the boundary connections. However, $Z_{\mathrm{PBC}}/Z_{\mathrm{OBC}} \approx 6 \gg 1$ (red) for the impedance across *all* inter-ladder pairs (rightmost plot), indicative of the non-local influence from boundary connections. (b) PBC and OBC impedance ratios extrapolated to larger system sizes, across various types of intervals: A and B nodes of the $\lceil \frac{N}{2} \rceil$th unit cell (Middle Inter), 1st unit cell (Edge Inter), 1st and $N$th unit cells (Edge-to-edge Inter); AA or BB nodes of the $\lfloor \frac{N}{2} \rfloor$th and $\lceil \frac{N}{2} \rceil$th unit cells (Middle Intra), 1st and 2nd unit cells (Edge Intra), 1st and $N$th unit cells (Edge-to-edge Intra). Evidently, $Z_{\mathrm{PBC}}/Z_{\mathrm{OBC}}$ is significantly higher than unity (dashed line) even for large $N$, further establishing that the boundary connections affect faraway impedances non-locally.

correspondence i.e. that perturbing a "boundary" coupling can significantly affect the spectrum of the entire lattice [6–9, 11, 69]. However, it has never been proven that the directly measurable *impedance response* also exhibits similar sensitivity, particularly when the directed amplification channels cancel. Below, we shall verify the affirmative by showing experimental data on its non-local current response.

### Non-local impedance response measurements

To probe non-local impedance response, we build a circuit represented by Laplacian $J(k)$ (Fig. 1c), and measure the two-point impedance $Z_{ij}$ between all sets of nodes $i, j$ for both periodic and open boundary conditions (PBCs and OBCs), as elaborated in the Methods section. We used $N = 8$ unit cells, with capacitors $C = 1nF$, inductors $L = 1mH$ and resistors $R = 5k\Omega$ and $r = 50k\Omega$, giving rise to dimensionless parameters $t = \omega RC = 5$ and $v = R/(2r) = 0.05$.

Under PBCs, the first and last unit cells of the ladder are connected in a translation-invariant manner; under OBCs, their disconnected connections are grounded. Going from PBCs to OBCs amount to the elimination of the two end-to-end connections, which would naively seem like a tiny perturbation in a long ladder with large number of unit cells $N$.

Yet, our experimental measurements reveal that changing the boundary connections indeed has a dramatic non-local impact: Fig. 2a shows the ratio between the measured PBC and OBC impedances $Z_{\mathrm{PBC}}$ and $Z_{\mathrm{OBC}}$ across all pairs of nodes on our fabricated $N = 8$ ladders. When the two nodes $i \in \{A\}$ and $j \in \{B\}$ belong to different sides of the ladder i.e. different sublattices, the ratio is substantially higher than unity (red pixels in the rightmost plot). In other words, removing the boundary connections between the 1st and 8th unit cells always significantly affect the two-point impedance across the ladder, even if the measurement is taken across two nodes that are furthest from the boundaries (i.e. 4A and 4B).

*Selectivity of non-local impedance response*

Interestingly, the boundary connections only affect the impedance non-locally across the circuit ladder, not across the nodes within the same ladder. As seen in the left two plots in Fig. 2a, the ratio $Z_{\mathrm{PBC}}/Z_{\mathrm{OBC}}$ is close to unity (light blue) if the two nodes are on the same ladder, except when the nodes are at the boundaries themselves. Intuitively, this is because in an impedance measurement between two nodes of the same ladder, most of the current takes the most direct route within that ladder, and is not affected by the hidden non-reciprocity in the inter-ladder couplings.

This selective non-local impedance response is maintained even if we extrapolate to much longer circuit chains, where the boundary couplings will ordinarily give rise effects that are even more negligible. As evident in Fig. 2b, as the chain length $N$ increases, the ratio $Z_{\mathrm{PBC}}/Z_{\mathrm{OBC}}$ of impedances across the middle of the ladders (red asterisked) remains significantly greater than unity. Since the middle unit cell is furthest from the boundaries, the fact that the presence(absence) of boundary connections can lead to significantly different $Z_{\mathrm{PBC}}(Z_{\mathrm{OBC}})$ of a long circuit is clear evidence of non-local response. By contrast, the PBC and OBC impedance between any two points within the same ladder converge towards each other in the large $N$ limit (blue, Intra), signifying the lack of non-local influence from the boundary couplings. Perhaps most surprisingly, the inter-ladder impedance across the same edge (red circled) or across different edges (red squared) are not as strongly affected by PBCs or OBCs as compared to that involving middle unit cell nodes, even though it is the edge nodes that are directly connected by the boundary couplings. This suggests that the hidden directed amplification is manifested as more strongly *far away*, rather than in the proximity of the OBC cutoff.

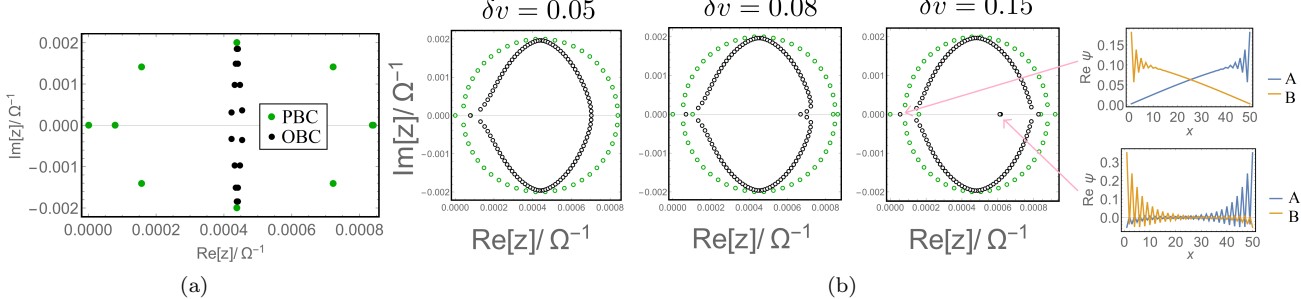

FIG. 3. **Laplacian spectrum of our circuit and emergent topological modes from parasitic resistances.** (a) The OBC and PBC Laplacian spectra of our circuit with $N = 8$ unit cells, which are very different due to the non-locality of boundary connections. Empirically measured parameters are $C = 1nF$, $L = 1mH$, $R = 5k\Omega$ and $r = 50k\Omega$, with parasitic resistances $R_{pc} = 2\Omega$ and $R_{pl} = 17\Omega$. (b) At a large ($N = 50$) system size, topological modes (isolated circles) emerges in our circuit as the coupling $\delta v = t^2(R_{pc} + R_{pl})/2R$ from parasitic resistances is increased: while a near-zero topological eigenvalue appears when $\delta v$ equals the coupling $v = 0.05$ from the resistors, an additional topological eigenvalue appears in the point gap beyond $\delta v = 0.08$. Their spatial profiles are manifestly boundary-localized, as presented for $\delta v = 0.15$ when the topological modes become well-separated from other eigenmodes.

### Reciprocal non-Hermitian skin effect

We would like to clarify the distinction between selective non-local impedance response and the breakdown of spectral bulk-boundary correspondence due to the reciprocal non-Hermitian skin effect [6, 11, 13, 20, 38]. Shown in Fig. 3a are the completely different PBC and OBC Laplacian spectra of our circuit, which indicates broken spectral bulk-boundary correspondence due to the "hidden" asymmetric couplings of a reciprocal system. However, having very different PBC and OBC spectra does not imply that impedance response is equally sensitive to the boundary connections. Explicitly, the impedance $Z_{ij}$ between two nodes $i$ and $j$ is related to the Laplacian eigenspectrum and eigenstates via[64]

$$Z_{ij} = \sum_{\mu} \frac{||\psi_{\mu}(i) - \psi_{\mu}(j)||^2}{z_{\mu}}. \quad (5)$$

Here $J\psi_{\mu} = z_{\mu}\psi_{\mu}$, $\mu = 1, ..., 2N$, and the biorthogonal norm [9, 15] is used since the Laplacian is non-Hermitian. Typically, $Z_{ij}$ is contributed by many $\mu$ terms of comparable magnitudes, and the spatial gradients $\psi_{\mu}(i) - \psi_{\mu}(j)$ of the eigenstates can conspire to produce approximately equal PBC and OBC $Z_{ij}$ even if their spectra $\{z_{\mu}\}$ are significantly different. Exceptions would be "topolectrical resonance" contributions from $z_{\mu} \approx 0$ eigenvalues - but note that in our case (Fig. 3a), the almost vanishing PBC $z_{\mu}$ arises from the arbitrariness of the reference voltage, and possesses an uniform eigenstate that does not contribute to the impedance.

### Size-dependent topological phase crossover from parasitic resistances

Usually, we expect inevitable parasitic resistances in the circuit components to erode experimental signatures, such that they must be minimized in order to have meaningful results. Yet, unexpectedly, parasitic resistances not only do not significantly threaten the non-local signatures in our fabricated circuit, but in fact stabilize enigmatic topological modes which appear when the OBC circuit is sufficiently long.

This unusual size-controlled topological phase crossover is a manifestation of the *critical* non-Hermitian skin effect [55], which has so far never been experimentally observed. Physically, it arises due to the highly non-linear scaling of the effective inter-chain couplings as $N$ increases. Even though the "bare" coupling is always $\Delta = -r^{-1}$, corresponding to a weak $v = R/2r = 0.05$ when put into dimensionless form, the exponentially large "virtual" non-Hermitian skin modes from the hidden directed amplification renormalizes their effective strengths to far larger values which are exponentially increasing with $N$. As such, we expect the same circuit to be in *different* regimes corresponding to effectively weak and strong coupling at small and large $N$ respectively.

The qualitative crossover between these two regimes occurs when the energetics within a single chain is comparable to the effective inter-chain coupling strength. For our circuit in particular, the effective coupling depends on the product of the bare coupling given by (see Eq. 10 in Methods)

$$v + \delta v \cos k = v + \frac{t^2(R_{pc} + R_{pl})}{2R} \cos k, \quad (6)$$

and a renormalization factor that increases rapidly with system size $N$. As $\delta v$ contributes to a sublattice modulation $\delta v \pm it$, it gives rise to Su-Schrieffer-Heeger (SSH)-like topological modes whenever the capacitors and inductors harbor sufficiently large parasitic resistances $R_{pc}$ and $R_{pl}$. Shown in Fig. 3b are the Laplacian spectra for three illustrative $\delta v$ at fixed $N = 50$. The two topological modes start to emerge at $\delta v = 0.05$ and $0.08$ respectively, and become more distinctively isolated and spatially well-defined at larger $\delta v$. In Fig. 8 in the Methods, the emergence of topological modes is documented across different system sizes

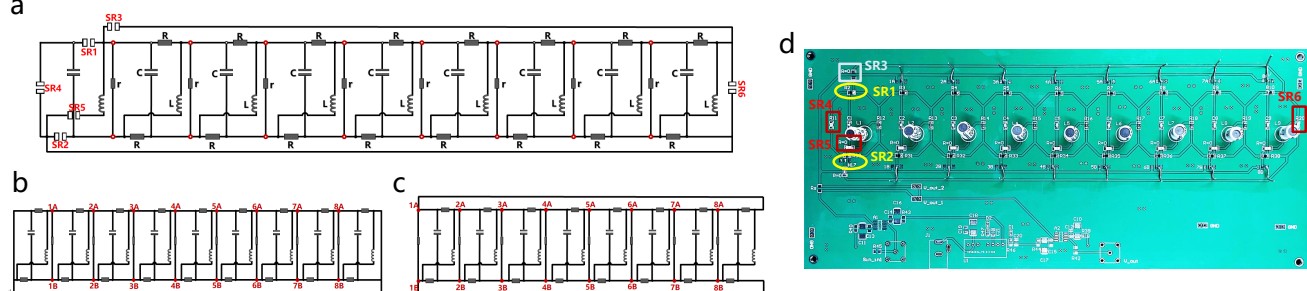

FIG. 4. **OBC and PBC switching in our circuit.** (a) Schematic of our circuit, with six gaps SR1 to SR6 that can be used to "switch" between open and periodic boundary conditions. (b) The OBC configuration is implemented by replacing SR1 and SR2 with resistors $R$, SR4, SR5 and SR6 by dissipationless wires and SR3 left disconnected. (c) The PBC configuration is implemented by replacing SR2 and SR3 with resistors $R$, and all the others left disconnected. (d) The full circuit board with the switches highlighted. Components used are rated $C = 1nF$, $L = 1mH$, $R = 5k\Omega$ and $r = 50k\Omega$.

$N$ for $\delta v = 0.1$, first appearing at $N = 20$ and becoming more distinct beyond that. In all cases, these size-controlled isolated modes only exist in the OBC and not PBC spectra, further confirming that they result from the critical non-Hermitian skin effect.

## DISCUSSION

Resistors, capacitors and inductors (RLC) are the simplest of electronic components, but yet in this work, we demonstrated theoretically and experimentally that they lead to enigmatic non-local impedance responses. While this was achieved through careful circuit design that appealed to a virtual directed amplification mechanism, qualitatively similar non-local responses should occur as long as non-reciprocity from LC phase delay is appropriately juxtaposed with resistive loss. Unique to electrical circuit platforms (and not photonic or acoustic media for instance), this non-locality in the impedance is distinct from the breakdown of spectral bulk-boundary correspondence from the reciprocal non-Hermitian skin effect, since the impedance between some pairs of nodes (such as our measured intra-ladder impedance) can remain almost unchanged by the removal of distant connections, even as the Laplacian spectrum has already been drastically modified.

As a robust non-Hermitian phenomenon, the non-local impedance response is comfortably robust against reasonable levels of parasitic resistances of the order of $\mathcal{O}(10)\%$. Perhaps surprisingly, such parasitic resistances in the capacitive and inductive elements even lead the fortuitous appearance of real topological modes at sufficiently large system sizes. These *loss-induced* topological modes emerges when the effect of time-reversal and (virtual) sublattice symmetry breaking from the parasitic resistive loss is compounded over many unit cells, and constitutes a fresh new manifestation of the critical skin effect.

Based purely on basic electrical circuit elements, our non-local mechanism is compatible with current technology for applications such as sensing[47–52]; its exclusive use of RLC elements make it area-friendly, functional stable and easily integrated into a chip. Besides, if we drop our lumped circuit assumption, our approach can be used to construct passive microwave circuits with non-local responses, with the promise of superior performance in impedance matching and tuning and as resonators, power dividers and directional couplers and filters compared to current microwave engineering designs. This would inspire new microwave technology from state-to-art physics [70, 71], and we leave it for future investigations.

## METHODS

### PBC/OBC circuit setup and measurements

Shown in Fig. 4a is the full schematic of our fabricated circuit with $N = 8$ unit cells. It assumes a ladder configuration, with resistors $R = 5k\Omega$ connecting successive nodes, resistors $r = 50k\Omega$ forming the rungs, and capacitors $C = 1nF$ as well as inductors $L = 1mH$ diagonally connecting adjacent nodes in the opposite rung.

To demonstrate non-local impedance responses, we design the circuit to be easily switchable between OBC and PBC configurations (Fig. 4b and c), such that 2-point impedance data under OBCs and PBCs can be readily compared. This is achieved through "switches" labeled SR1 to SR6 in the schematic as well as the photograph of the printed circuit board (Fig. 4d).

To implement OBCs or PBCs, SR1 to SR6 are to be substituted with resistors $R$, wires of negligible resistance, or simply left empty. For OBCs, SR1 and SR2 (yellow) are replaced by resistors $R$, and SR3 (white) is left disconnected. SR4, SR5 and SR6 (red) are replaced by dissipationless wires. This grounds the edge unit cells, yielding the OBC configuration of Fig. 4b. For PBCs (Fig. 4c), SR2 and SR3 are replaced by dissipationless wires, and SR1, SR4, SR5 and SR6 are disconnected. This restores the boundary connections of the PBC configuration of Fig. 4c.

Since the exact resonance frequency $\omega_0$ is not a priori known due to component uncertainty, we sweep through the relevant AC frequency range and identify $\omega_0$ from the impedance peak. The two-point impedance between any two nodes is measured by connecting an LCR meter across the nodes.

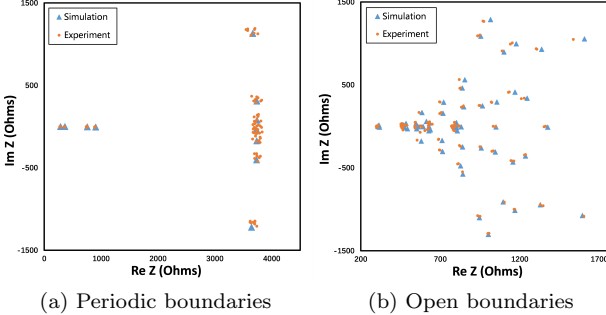

(a) Periodic boundaries    (b) Open boundaries

FIG. 5. **Empirical determination of parasitic resistances.** Experimentally measured impedances across all $2N(N-1)/2 + N = 8 \times 7 \times 2/2 + 8 = 64$ unique pairs of nodes (orange) are compared with the impedances computed from Eq. 10 (blue), with effective parasitic resistances adjusted to $R_{pc} = 2\Omega$, $R_{pl} = 17\Omega$ such that the fit is optimal.

Circuit simulations were performed with the Cadance virtuoso software. A sinusoidal signal with magnitude 1V was connected across nodes to measure their 2-point impedance, with the resonant AC frequency $f_0 = 2\pi\omega_0$ determined by searching across the range $150kHz \sim 170kHz$.

*Parasitic resistances and accurate modeling of our circuit Laplacian*

To accurately model our circuit, it is necessary to assume some level of parasitic resistance in each capacitor

and inductor. Through measurements on single components, it was found that using capacitors of $C = 1nF$ and inductors of $L = 1mH$ minimizes the effects of parasitic resistances while keeping the resonance frequency in the order of $\omega_0 = (LC)^{-1/2} = 10^6 s^{-1}$, which is convenient for measurements. By adding serial parasitic resistances to them in our simulations, and comparing the simulated and experimentally measured impedances across all pairs of nodes (Fig. 5), we found that it suffices to assume a common serial parasitic resistance $R_{pc}$ and $R_{pl}$ to all capacitors and inductors respectively:

$$i\omega C \to \frac{i\omega C}{1 + i\omega C R_{pc}} \tag{7}$$

$$\frac{1}{i\omega L} \to \frac{1}{R_{pl} + i\omega L} \tag{8}$$

where $R_{pc} = 2\Omega$ and $R_{pl} = 17\Omega$. These parasitic resistance values optimize the fit between the experimental and simulated impedances, with magnitude and argument discrepancies respectively smaller than 4% and 2% respectively for most data points. To recall, the other component parameters are $C = 1nF$, $L = 1mH$, $R = 5k\Omega$, $r = 50k\Omega$, such that the resonance frequency is $f_0 = \omega_0/2\pi = 159.15kHz$, and $t = \omega_0 RC = 5$, $v = R/(2r) = 0.05$. Note that with these empirically parameters, the parasitic corrections to the $C$ and $L$ are very small, of the order of 0.2% and 1.7% respectively.

Substituting Eq. 8 into Eq. 2 of the main text, we arrive at the experimental circuit Laplacian

$$J_{\exp}(k) = \frac{i\omega C}{1 + i\omega C R_{pc}} \begin{pmatrix} 1 & -e^{ik} \\ -e^{-ik} & 1 \end{pmatrix} + \frac{1}{R_{pl} + i\omega L} \begin{pmatrix} 1 & -e^{-ik} \\ -e^{ik} & 1 \end{pmatrix} + \frac{1}{r} \begin{pmatrix} 1 & -1 \\ -1 & 1 \end{pmatrix} + \frac{2 - 2\cos k}{R} \begin{pmatrix} 1 & 0 \\ 0 & 1 \end{pmatrix}. \tag{9}$$

At resonance, such that $\omega = \omega_0 = \frac{1}{\sqrt{LC}} = 10^6$,

$$
\begin{aligned}
J_{\exp}(k)|_{\omega=\omega_0} &= \frac{i\omega_0 C}{1 + i\omega_0 C R_{pc}}(\mathbb{I} - \cos k\sigma_x + \sin k\sigma_y) + \frac{1}{R_{pl} + i\omega_0 L}(\mathbb{I} - \cos k\sigma_x - \sin k\sigma_y) + \frac{1}{r}(\mathbb{I} - \sigma_x) + \frac{2 - 2\cos k}{R}\mathbb{I} \\
&\approx \frac{2}{R}[i(t\sin k)\sigma_y + v(\mathbb{I} - \sigma_x) + (1 - \cos k)\mathbb{I}] + i\omega_0 C\left(\frac{1}{1 + i\omega_0 R_{pc}C} - \frac{1}{1 + \frac{R_{pl}}{i\omega_0 L}}\right)[\mathbb{I} - \sigma_x\cos k] \\
&\approx \frac{2}{R}[i(t\sin k)\sigma_y + v(\mathbb{I} - \sigma_x) + (1 - \cos k)\mathbb{I}] + (\omega_0 C)^2(R_{pc} + R_{pl})[\mathbb{I} - \sigma_x\cos k] \\
&= \frac{2}{R}\left[i(t\sin k)\sigma_y + v(\mathbb{I} - \sigma_x) + (1 - \cos k)\mathbb{I} + \frac{t^2(R_{pc} + R_{pl})}{2R}(\mathbb{I} - \sigma_x\cos k)\right], \tag{10}
\end{aligned}
$$

where $t = \omega_0 RC = R\sqrt{\frac{C}{L}} = 5$ and $v = R/2r = 0.05$. Going from the 1st to the 2nd line, we made the approximation that $i\omega_0 C/(1 + i\omega_0 C R_{pc})^{-1} + (R_{pl} + i\omega L)^{-1} \approx i\omega_0 C + (i\omega L)^{-1}$, which holds because $\omega_0 R_{pc}C \ll 1$ and $R_{pl}/i\omega_0 L \ll 1$. This is also used to simplify line 3 from

line 2.

Comparing with Eq. 3 of the main text, the additional term from the parasitic resistances is the rightmost term containing $(\mathbb{I} - \sigma_x\cos k)$, with coefficient $\frac{t^2(R_{pc}+R_{pl})}{2R} \approx 0.0475$ that is comparable in magnitude

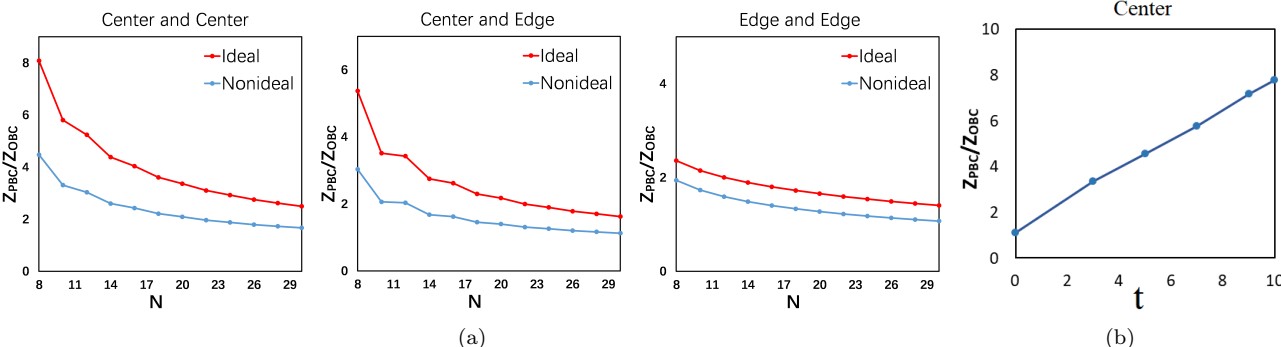

FIG. 6. **Scaling of impedance ratio $Z_{PBC}/Z_{OBC}$ against $N$ and $t$.** (a) Simulation results of $Z_{PBC}/Z_{OBC}$ vs $N$, $N = 8, 10, ..., 30$, comparing between ideal cases without parasitic resistances, and non-ideal cases with parasitic resistances empirically obtained as $R_{pc} = 2\Omega$ and $R_{pl} = 17\Omega$. While this inter-ladder impedance ratio is consistently lower for the non-ideal case, the center-to-center interval still consistently exhibits a ratio larger than unity, signifying response non-locality. (b) Center-to-center $Z_{PBC}/Z_{OBC}$ at $N = 8$ (across the A and B nodes of the 4th unit cell) and its almost linear dependence on $t = \omega RC \approx RC/\sqrt{LC} = R/1000$. $v = R/2r = 0.05$ is kept constant.

with the other symmetric ($\sigma_x$) inter-ladder coupling $v$.

### Scaling behavior of circuit

#### *Extrapolation to longer circuit ladders*

Here, we present further results on the scaling behavior of the two-point impedance. As shown in Fig. 6a, the inter-ladder impedance ratio $Z_{\mathrm{PBC}}/Z_{\mathrm{OBC}}$ decreases universally as the system size $N$ increases. However, the ratio does not converge to unity when the two points are at the center of the ladders, but to an asymptotic value that is significantly larger than one, implying the robustness of the non-local response even with the empirically extracted parasitic resistances included ("non-ideal" case). In general for inter-ladder intervals, larger $Z_{\mathrm{PBC}}/Z_{\mathrm{OBC}}$ ratios exist either across nearby points, or if one or both of the points are near the edge (Fig. 7).

Interestingly, at the fixed system size of $N = 8$, the non-local response is almost linearly proportional to $t = \omega_0 RC = R\sqrt{C/L}$, as shown in Fig. 6b. This is because $t$ can be interpreted as the "hidden" coupling asymmetry of the effective two HN-chain model. In the limit of large $t$ i.e. large $C$ and/or small $L$, we expect a vastly larger $Z_{\mathrm{PBC}}$ vs. $Z_{\mathrm{OBC}}$ across the ladders, and that limit could be further employed to generate very strong non-local response.

#### *Emergent topological mode at large system sizes*

Here, we provide more detailed plots of the Laplacian spectrum of our circuit as $N$ increases, such as to substantiate our discussion of the size-dependent topological phase crossover. In Fig. 8, the emergence of two real topological modes - one near 0 and the other in the point gap - is clearly observed as $N$ increases from 20 to 150. This provides further evidence that

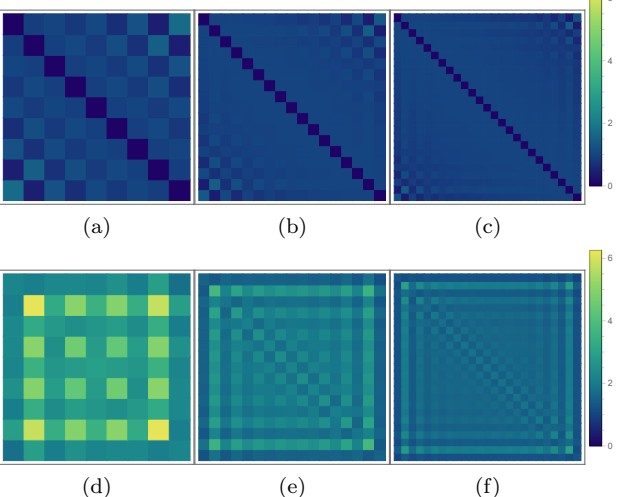

FIG. 7. **Two-point impedance ratios $Z_{\mathrm{PBC}}/Z_{\mathrm{OBC}}$ in longer circuit arrays.** (a-c) Intra-ladder impedance ratios for $N = 9, 17$ and 25 respectively. (d-f) Inter-ladder impedance ratios also for $N = 9, 17$ and 25 respectively. The circuit dimensionless parameters are $t = 5$, $v = 0.05$ and $\delta v = 0.0475$ as obtained from our experimental setup.

parasitic losses in our circuit gives rise to the critical non-Hermitian skin effect in the form of a topological crossover.

*Data availability* – The data that support the findings of this study are available from the corresponding author upon reasonable request.

### END NOTES

*Acknowledgements* – X.Z. is supported by the National Natural Science Foundation of China (Grant No. 11874431), the National Key R&D Program

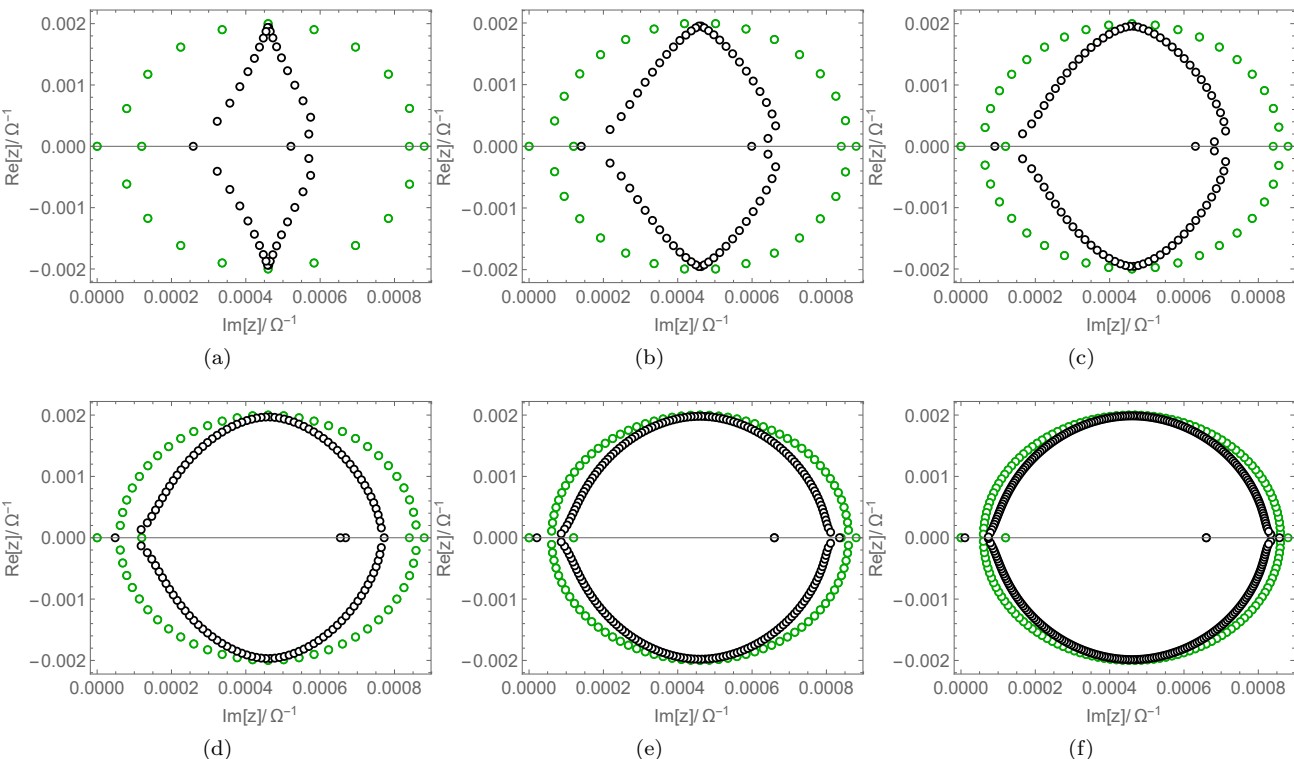

FIG. 8. **Emergence of topological modes as system size is increased.** (a-f) Shown are the PBC (green) and OBC (black) spectra for $\delta v = 0.1$ and system sizes $N = 20, 30, 40, 60, 100$ and $150$ respectively. Other parameters are set at $t = 5$ and $v = 0.05$ as in our experiment. Budding topological modes (isolated small black circles) start to emerge at $N = 20$ along the real line, and become more and more isolated and distinct as $N$ increases. Beyond $N = 60$, one of them gravitates towards 0, and the other is well within the point gap of the spectrum.

of China (Grant No. 2018YFA0306800), and the Guangdong Science and Technology Innovation Youth Talent Program (Grant No. 2016TQ03X688). C.H.L acknowledges support from Singapore's Ministry of Education Tier-1 Grant A-8000022-00-00.

**Correspondence** and requests for materials should be addressed to C.H.L. and X.Z.

The authors declare no competing interests.

*Author contributions* –C.H.L conceptualized the idea, supervised the project and wrote the manuscript. C.H.L. and X.Z. designed the experiment, and interpreted the results. X.Z., B.X.Z. and W.H.Z. performed the simulation, fabricated the circuit and performed the experiemnt.

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
