# Peer review of "Observation of non-local impedance response in a passive electrical circuit"

_SciPost Physics_

## Round 1 · Referee Report · Anonymous (Referee 1) · 2023-7-20

Strengths

  1. This research is on a topic which is very important and is being very actively studied in recent years.
  2. The theoretical predictions are corroborated by real experiments.
  3. This work is nicely put into the context of quite extensive literature and variety of results from different research groups.

Weaknesses

  1. I found that the interpretation of the main result, given in the subsection "Size-dependent topological phase crossover..." lacks physical clarity and rigor. The feature they found apparently is not a continuous /discontinuous transition, or disorder line. It is a finite-size effect, and I don't think that one can call it "a crossover".

Report

The manuscript presents results on a very interesting and physically highly relevant topic of simulating topological quantum materials and transitions in electrical circuits. The text in general is well written and structured. Other contributions in this very active and competitive field are properly cited and acknowledged. The manuscript in my opinion meets at least one acceptance criterion which must be satisfied, according to the journal guidelines: "(3) - open a pathway in an existing or a new research direction...."
I have however one concern regarding the "topological phase crossover" (or "critical non-Hermitian skin effect"). The nature of this phenomenon needs to be clarified and explained in a self-contained form in the manuscript. As discussed in Ref. 55, it can be related to the behavior of energy zeros on the complex plane of wave numbers $z=e^{ik}$. Is it a disorder line of the first or second kind? (According to the definitions of Stephenson, PRB 1, 4405 (1970)). If it is a finite-size effect, where are the parametric boundaries for it to disappear and how it is related to the positions of zeros $z$ on the complex plane?
I think this issue needs to be addressed before the manuscript can be recommended for publication.

Requested changes

  1. Subsection "Size-dependent topological phase crossover..." needs to be revised as discussed above.
  2. Consequently, some related comments on the physical nature of the phenomenon in question would improve the Introduction and Conclusion.

---

## Round 1 · Referee Report · Anonymous (Referee 2) · 2023-8-4

Strengths

Experimental observation of non-local impedance response in a passive electrical circuit

Weaknesses

Lack of clarity what are experimental findings and which ones are theory (simulation)

Report

This work investigates the impedance response of an L-R-C circuit that realizes the geometry of a two-leg ladder with both open (OBC) and periodic boundary conditions (PBC) along the legs. A non-local response is found in the cross-leg impedance response ratio $Z_{\rm PBC}/Z_{\rm OBC}$.

I understand that the authors have fabricated an $N=8$ device and performed complementary simulations for different, in particular larger values of $N$. However, it is not always clear from the text which results are experimental findings and which ones obtained by simulation. I believe that this point needs clarification.

On another note, the manuscript seems to have been prepared for a different target journal, and I think that it would be useful if the authors could reformat it according to the SciPost Physics guidelines, see https://scipost.org/SciPostPhys/authoring. In particular, the "Methods" sections should be either integrated into the main text (no length limitation) or moved to Appendices.

I have one question: the values $C=1\,{\rm nF}$, $L=1\,{\rm mH}$, $R=5\,{\rm k \Omega}$, $r=50\,{\rm k\Omega}$ are nominal values (?). Have the authors verified if their components agree with the specifications? In view of parasitic resistances on the order of 0.2 to 1.7%, one might imagine deviations from nominal values on the same order of magnitude.

Further more specific items are listed as "Requested changes".

Requested changes

1- State clearly which results are experimental ones and which one come from simulations. A comparison between the two may also be appropriate. 2- Reformat the manuscript according to SciPost Physics guidelines, https://scipost.org/SciPostPhys/authoring. 3- Clarify if the values for $C$, $L$, $R$, and $r$ are nominal ones, or if the components actually have these values. 4- First paragraph of the Introduction: The list of 33 references [6-38] is not very helpful to the reader. The authors should either break this down into smaller units with appropriate comments, or reduce the list to really relevant references. 5- At the bottom left of page 2, the authors say that the Laplacian $J$ relates ${\bf V}$ to ${\bf I}$, but then they write the opposite equation ${\bf I} = J{\bf V}$. As long as $J$ is invertible, the two are of course equivalent, but presentation should nevertheless be coherent. 6- Format Eq. (4) properly, i.e., it should start with a new paragraph, not in line with the text. 7- Some figures use fonts that are too small. This applies in particular to Figs. 2, 5, and 7, and their legends. 8- On page 4, there is a discussion of "parasitic" resistance, but this is not properly explained before page 6 (one point where the manuscript would benefit from restructuring). 9- Units are missing after the $10^6$ below Eq. (9) (${\rm Hz}$?). 10- Caption of Fig. 6: the definition of $t$ is hidden somewhere in the text. For clarity, I recommend to recall it here. 11- Update references, specifically: [29] Science Bulletin 67, 1865-1873 (2022) [30] Phys. Rev. B 106, 075158 (2022) [34] Phys. Rev. A 107, L010202 (2023) [35] Phys. Rev. B 107, L220301 (2023) [61] Phys. Rev. Research 5, L012041 (2023) [70] Nature Photonics, 120–125 (2023) ... and add DOIs.

---

## Editorial Decision

resubmitted